# Ext4 and XFS File System Forensic Framework Based on TSK

**Hyungchan Kim** [1,2], **Sungbum Kim** [1], **Yeonghun Shin** [1], **Wooyeon Jo** [3], **Seokjun Lee** [4] **and Taeshik Shon** [1,5,*]

1 Department of Artificial Intelligence Convergence Network, Ajou University, Suwon 16499, Korea;
hj1003hj@ajou.ac.kr (H.K.); zx1962@ajou.ac.kr (S.K.); syh2347@ajou.ac.kr (Y.S.)
2 Platform Tech Team, WINS Co., Ltd., Seongnam-si 13487, Korea
3 Department of Computer Engineering, Ajou University, Suwon 16499, Korea; dndusdndus12@gmail.com
4 Department of Computer Science, Kennesaw State University, Marietta, GA 30060, USA;
slee235@kennesaw.edu
5 Department of Cyber Security, Ajou University, Suwon 16499, Korea
* Correspondence: tsshon@ajou.ac.kr

**Abstract:** Recently, the number of Internet of Things (IoT) devices, such as artificial intelligence (AI) speakers and smartwatches, using a Linux-based file system has increased. Moreover, these devices are connected to the Internet and generate vast amounts of data. To efficiently manage these generated data and improve the processing speed, the function is improved by updating the file system version or using new file systems, such as an Extended File System (XFS), B-tree file system (Btrfs), or Flash-Friendly File System (F2FS). However, in the process of updating the existing file system, the metadata structure may be changed or the analysis of the newly released file system may be insufficient, making it impossible for existing commercial tools to extract and restore deleted files. In an actual forensic investigation, when deleted files become unrecoverable, important clues may be missed, making it difficult to identify the culprit. Accordingly, a framework for extracting and recovering files based on The Sleuth Kit (TSK) is proposed by deriving the metadata changed in Ext4 file system journal checksum v3 and XFS file system v5. Thereafter, by comparing the accuracy and recovery rate of the proposed framework with existing commercial tools using the experimental dataset, we conclude that sustained research on file systems should be conducted from the perspective of forensics.

**Keywords:** file system; digital forensic; file recovery; digital investigation; The Sleuth Kit

## 1. Introduction

The Linux file system is used in various operating systems, such as Debian, Red Hat, and Fedora, and uses Ext as the default file system. In addition, most Android smartphones and Internet of Things devices (e.g., artificial intelligence speakers) use Ext as the basic file system [1–3]. Recently, Linux file systems, such as XFS, the Open Zettabyte file system (OpenZFS), and Btrfs, have been released for large files and with improved processing speed and scalability. Red Hat 7.0, BUFFALO's NAS products, and Samsung Techwin's CCTV products use XFS as the default file system. The user and system information, usage traces, files, etc., are generally stored in the file system, and data may be deleted due to unexpected errors, accidents, or anti-forensics.

The deleted data in the file system may have significant value from the point of view of digital forensics; hence, the method of extracting and recovering files stored in the file system has been actively studied. In some studies, a file extraction and recovery method based on the open metadata of the Ext2/3 file system [4] and a deleted file recovery method based on the journal area of the Ext4 file system [5,6] have been suggested. Moreover, studies on the metadata analysis of the XFS file system and the recovery method of deleted files have been implemented [7–9]. Most of the studies on file extraction and recovery methods for file systems are performed based on metadata and journal areas; hence, an accurate analysis of the structure of the file system is required. Due to the file system

characteristics, the structure of the file system may change due to periodic version updates for security and stability. Accordingly, metadata and journal structure analysis for each version of the file system must be performed.

Based on existing research on file systems, commercial tools and open sources that can be used in an actual investigation have been developed. For example, TSK [10,11], an open-source tool developed in 2005, provides metadata-based extraction and recovery of file systems. It supports major file systems, such as the New Technology File System (NTFS), File Allocation Table (FAT), Extended File System (Ext), Hierarchical File System (HFS), Unix File System (UFS), and Apple File System (APFS). In addition, it provides command line-based file system forensic functions, such as fsstat, which outputs the metadata of supported file systems, and tsk_recover, which performs file extraction and recovery. It is used in actual digital forensic investigations by various investigative agencies. Currently, through periodic version updates, TSK developers are expanding the supported file system by rectifying errors that occur in some environments of the file system. Nevertheless, TSK cannot recover deleted files because it does not provide a recovery function using the journal area of the Ext4 file system. Moreover, it does not support the latest file system (i.e., XFS); consequently, data extraction and recovery cannot be performed. The inability to recover necessary data during an actual forensic investigation can lead to serious consequences; hence, forensic research on a continuous file system is critical. The contributions of this paper are as follows.

- The characteristics of the journal area used for the recovery of deleted data in the Ext4 file system and those of the metadata used for data extraction in the XFS file system are derived.
- A file system forensic tool capable of recovering deleted data from the Ext4 file system and extracting XFS file system data based on TSK using the characteristics of each derived file system is developed.
- The developed TSK-based file system forensic tool can be applied to large-capacity environments exceeding 100 GB.

This paper is organized as follows. Section 2 presents the research related to data extraction and recovery in the file system. Section 3 discusses information recovery using the proposed TSK-framework-based Ext4 journal area and data extraction and recovery using the XFS metadata. Section 4 explicates the evaluation of the tools presented in this paper, and Section 5 elaborates on the experimental results. Finally, Section 6 summarizes the conclusions and future work.

## 2. Related Work

This section deals with the description and limitations of the existing Ext file system and XFS file system metadata analysis, deleted data recovery methodology, and TSK-based file system forensics research.

### 2.1. Ext File System Structure Analysis and Recovery Method

In 2012, an analytical study on Ext4 was conducted from the perspective of digital forensics using journaling techniques [5]. A detailed analysis was performed on the structure of Ext4, and a recovery method considering the compatibility with Ext3 was proposed. However, complete file recovery is difficult to comprehend because the method only analyzes the metadata of the Ext4 file system and does not indicate objective results, such as recovery rate.

In 2014, a file recovery that can be used even when the block pointer for the Ext2/3 file system is deleted was proposed [4]. Unfortunately, the default file system of the latest Linux kernel is Ext4, which uses the extent method instead of the block mapping method. This method, which can recover deleted files only in a limited environment (i.e., Ext2/3 file system), is difficult to apply because it is not suitable for the Ext4 file system.

In 2019, a study on file recovery for the Ext4 file system analyzed the journal area, indexed metadata related to deleted files, and proposed a recovery method based on these

metadata [6]. A deleted file recovery tool was also implemented. Through performance comparison with other commercial tools, it was demonstrated to be a specialized technique for large file systems. However, major updates, such as the checksum area, have been introduced to the Ext4 file system journal area. As a result, the formulated tool cannot be applied to this file system at this time. Currently, Ext4 uses journal checksum v3; hence, a new recovery method according to the updated metadata is necessary.

*2.2. XFS File System Structure Analysis and Recovery Method*

Since 2014, forensic research based on the XFS file system has been conducted. Most of the file system structures have been analyzed, and recovery methods for deleted files have been suggested. In 2014, a study on the structure of the XFS file system and recovery of deleted files was conducted based on XFS file system v4. For the recovery, the inode of the deleted file was located, and the address of the bitmap block or its signature was used [7]. However, this recovery method cannot recover a deleted file if the bitmap block value has been overwritten.

To compensate for the aforementioned case, a file recovery method using the journal area of the XFS file system has been examined [8]. The recovery of a deleted file is achieved by leveraging the feature that information, such as the size and offset of the deleted file, remains in the journal area of the XFS file system. However, it is very difficult to apply because the experimental environment that can affect the file system metadata is not detailed, and the current XFS version v5 is newer than tested. In the case of UFS Explorer, which is a commercial tool, it has been found that it is impossible to recover files less than 1 KB compared to supporting the recovery of files having 3 GB or more [9]. Since UFS explorer is not open-source, it is difficult to accurately identify the cause of poor performance, but it can be inferred that metadata are not properly utilized.

*2.3. TSK-Based File System Extraction and Recovery*

TSK is an open-source file system forensic tool based on the data extraction method using the metadata of the Linux file system proposed by Brian Carrier in 2005 [11]. Although this tool provides a wide range of functions targeting various file systems, TSK has a limitation: it encounters problems in supporting the latest file systems. Accordingly, using the advantages of the open-source project, file system forensic researchers have conducted studies to overcome the limitations of TSK. In 2017 and 2018, the data structures of ZFS and Btrfs were analyzed, and a data extraction tool based on TSK was implemented [12,13]. Both papers proposed a data extraction model based on pool storage, which was also used in this work to extract data. However, the foregoing model did not properly follow the development framework of TSK and only employed certain functions, resulting in insufficient extensibility. It did not even implement the recovery of deleted files; hence, practicality was not sufficiently secured to be used for actual investigation.

## 3. Proposed File System Forensic Framework

The file extraction and recovery tools developed for specific versions of file systems may not work normally if the file system version has been updated. This is because the metadata structure employed for file extraction and recovery has been changed by the update. In particular, TSK, a well-known file system forensic tool, does not consider the journal area in the Ext4 file system and has no function for recovering deleted files. In the case of the XFS file system, file extraction and recovery are impossible because it is not supported by TSK. This section presents the analysis of the Ext4 file system journal checksum v3 and XFS file system v5 to identify the modified metadata structures and propose a file extraction and recovery framework based on TSK.

The proposed framework is based on TSK, which is well-known to users as open-source digital forensics and is easily extensible. The TSK-based framework architecture for recovering deleted files from an Ext4 file system and extracting files from an XFS file system is shown in Figure 1. The proposed framework operates based on the file extraction

and recovery command in TSK (i.e., tsk_recover); it does not affect other TSK functions. The proposed file recovery method for the Ext4 file system and the file extraction method for the XFS file system are as follows.

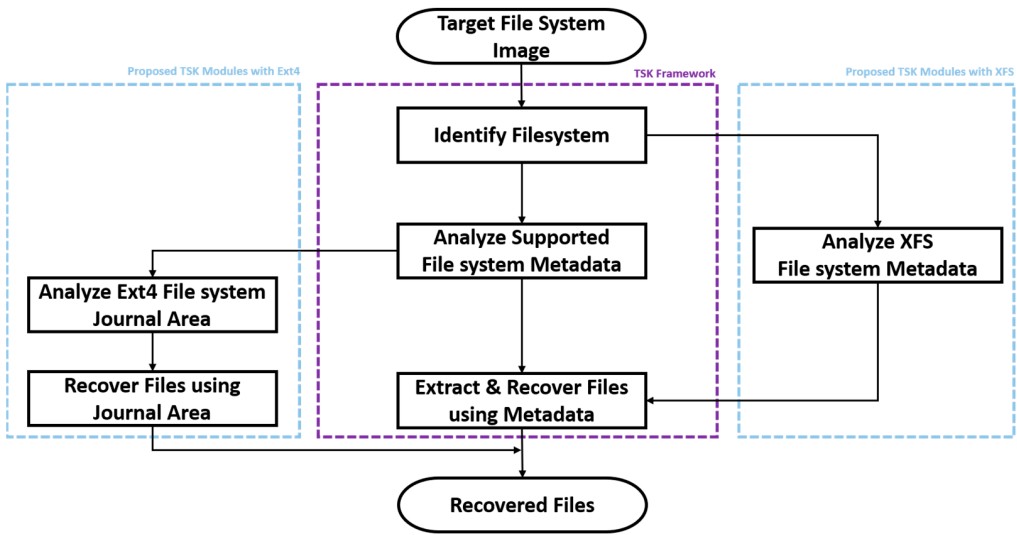

**Figure 1.** Ext4/XFS file system file recovery framework based on TSK.

- Ext4 journal-based file recovery method: Based on TSK's metadata analysis and recovery process, we additionally implement the journal area analysis and recovery function. The function can recover deleted files without affecting other TSK-Ext4 functions.
- XFS v5-based file extraction method: The XFS file system is not supported by TSK; hence, we implement a metadata structure analysis based on TSK. With this implementation, the extraction of existing files in the XFS is possible.

### 3.1. Ext4 Journal-Based TSK File System Forensic Module

To preclude system crash or blackout, Ext4 file systems use a journal area. In general, Ext4 uses ordered mode journaling, which records the metadata in the journal area when a file changes. Thus, the file can be restored to its previous state using the journal area if a problem is encountered when the file is modified. Lee et al. [6] analyzed the journal area of Ext4 and found that the metadata of deleted files are backed up in the journal. Then, they proposed a method for recovering deleted files using the journal area. Similarly, in this section, a TSK-based framework for file recovery [6] using the Ext4 journal is proposed.

#### 3.1.1. Ext4 File System Metadata for File Recovery

The metadata used for the file extraction and recovery of the Ext4 file system include the inode, directory entry, and journal. In TSK, although metadata analysis is performed, journal area analysis for the recovery of deleted data is not implemented. Accordingly, this section focuses on file recovery using the journal area. The journal area is basically allocated to inode 8, as may be checked in the superblock of Ext4. It consists of the journal superblock, journal descriptor block, journal data block, revocation block, commit block, and N transactions. Each transaction is composed of a journal descriptor block, journal data block, revocation block, and commit block. The journal metadata used for file recovery include the journal superblock, journal descriptor block, and journal data block.

- Journal Superblock: Based on Figure 2, the journal superblock contains basic information on the journal and starting position of the transaction log. It also includes a journal block header and starts with a magic number, 0xC03B3998. The journal block header is at the beginning of every journal block, and the value of the entry type field identifies the block.

- Descriptor Block: The descriptor block consists of an array of journal block tags that describe the location of the data block (Figure 3). The journal block tag has a different structure depending on the value of the s_feature_incompat flag of the journal superblock. If this flag value is set to JBD2_FEATURE_INCOMPAT_CSUM_V3 (0 × 10), then it basically has the same structure as ext2fs_journ_dentry_V3; otherwise, it has the same structure as ext2fs_journ_dentry_V2. The t_blocknr field of the journal block tag indicates the location where the data block is stored on the disk, implying that the inode of the deleted file is backed up in the journal.

| | 0 | 1 | 2 | 3 | 4 | 5 | 6 | 7 | 8 | 9 | A | B | C | D | E | F |
|---|---|---|---|---|---|---|---|---|---|---|---|---|---|---|---|---|
| 0x00 | Magic Number (0xC03B3998) | | | | Entry Type | | | | Entry Seq | | | | Block Size | | | |
| 0x10 | Number of blocks in journal area | | | | First Block of log information | | | | First commit ID in log | | | | Block number of the start of log | | | |
| 0x20 | Error value | | | | s_feature_compat (Checksums in the data blocks) | | | | s_feature_incompat | | | | s_feature_ro_compat | | | |

**Figure 2.** Journal superblock and journal block header of Ext4 file system.

ext2fs_journ_dentry_V2

| A | 0 | 1 | 2 | 3 | 4 | 5 | 6 | 7 | 8 | 9 | A | B | C | D | E | F |
|---|---|---|---|---|---|---|---|---|---|---|---|---|---|---|---|---|
| 0x00 | Common header identifying | | | | Checksum of the UUID | | Block Flag | | t_blocknr_high (where that data block should end on disk) | | | | UUID | | | |

ext2fs_journ_dentry_V3

| A | 0 | 1 | 2 | 3 | 4 | 5 | 6 | 7 | 8 | 9 | A | B | C | D | E | F |
|---|---|---|---|---|---|---|---|---|---|---|---|---|---|---|---|---|
| 0x00 | Common header identifying | | | | Block Flag | | | | t_blocknr_high (where that data block should end on disk) | | | | Checksum of the UUID | | | |

**Figure 3.** Journal descript block and journal block tag structure of Ext4 file system.

### 3.1.2. Ext4 Journal-Based File Recovery Method

The deleted file recovery framework using the Ext4 journal area proposed in this paper is shown in Figure 4. First, the framework finds the field value of s_feature_incompat in the journal superblock to recover deleted files. The journal block tag structure of the journal descriptor block varies depending on this value. When the field values of s_feature_incompat are set to 1 and 2, the structures are ext2fs_journ_dentry_V2 and ext2fs_journ_dentry_V3, respectively.

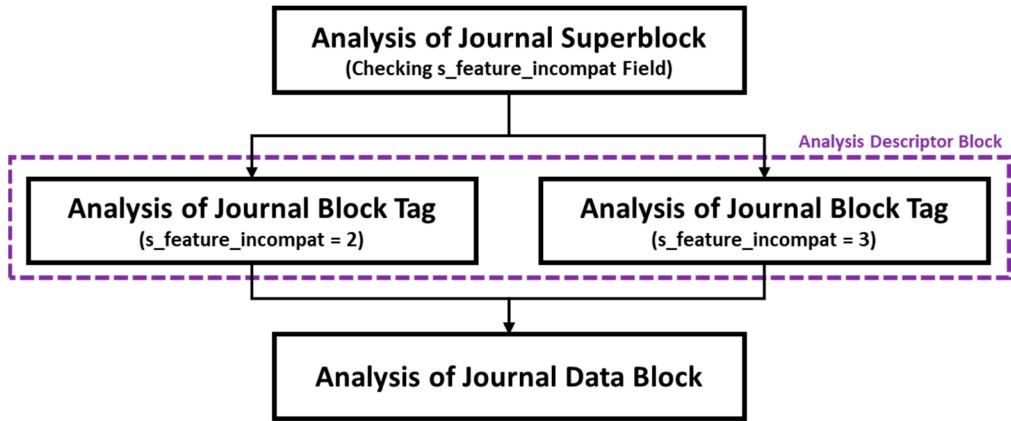

**Figure 4.** Ext4 journal-based file recovery module using TSK.

Then, the journal data block can be found by analyzing the journal block tag of all journal descriptor blocks existing in the journal area with the block number of the deleted inode. The journal data block stores the backed-up inodes of the deleted data, making the recovery of deleted files possible. If multiple backups of the same inode are stored in the journal, then the most recent inode backup with the a_time field value is recovered.

Metadata and journal analyses must precede file recovery from the Ext4 file system. The metadata analysis function of Ext4 is already implemented in the framework of TSK. Therefore, in the proposed Ext4 recovery module, the journal analysis function is additionally implemented. Then, based on the analyzed information, each inode of Ext4

is read sequentially. In analyzing the inode, the module checks whether the file pointed to by the inode has been deleted. If this is the case, then it extracts the deleted file information. If the file deletion time of the inode (d_time value) is not 0, this means that the file has been deleted; this operation is performed through the existing tsk function that sequentially reads the inode of Ext4. If the file has been deleted, then the inode is recovered from the journal. If the journal block tag and block address to which the inode of the deleted file belongs are the same, then the inode of the deleted file can be found in the journal data block. By implementing this process in the file extraction framework of TSK, deleted files can be recovered.

### 3.2. XFS v5-Based TSK File System Forensic Module

In the latest Linux version, which is supported by the xfs-progs 3.2.0-alpha1 version updated in July 2014, an XFS file system is typically created as an XFS file system v5 by default. In the case of XFS file system v5, applying the file extraction and recovery method performed in the XFS file system v4 is difficult because the metadata have been changed. In addition, studies on metadata structure analysis as well as file extraction and recovery for XFS file system v5 have not been conducted.

### 3.2.1. XFS File System Metadata for File Extraction

The main metadata used for the file extraction from the XFS file system include the superblock, inode, directory entries, and data extents. Detailed descriptions are as follows.

- Superblock: The superblock contains the general information of the file system; its structure is presented in Figure 5. The superblock stores the magic number, block size, inode size, sector size, etc., of the XFS file system. In XFS file system v5, fields have been added to ensure the integrity of the superblock crc and metadata uid.

- Inode list: The inode list stores the attribute information of the file or directory mapped to the inode and the location of the actual file. It is composed of the inode core and inode data fork. In XFS file system v5, fields to ensure integrity, creation time, original inode, etc., have been added, allowing the file information to be expressed in more detail; the size has been increased from 96 bytes to 176 bytes. The inode list address can be found using the inode number, whose structure is shown in Figure 6.

- Inode Core: The inode core contains attribute information of files or directories mapped to inodes; its structure is summarized in Figure 7. The inode data fork structure corresponding to the value of the di_format field appears behind the inode core.

- Directory Entries: Directory entries contain the name and inode offset of a file in the directory; its structure is summarized in Figure 8. If the di_format field of the inode core is 1, then the inode data fork consists of a structure of directory entries. Directory entries are created in the shortform, block, node, leaf, or B+tree entry form according to the number of files included in the directory and directory depth. In general, directory entries are created in shortform if the directory contains no more than 256 files.

- Data Extents: Data extents contain data related to the storage location of the file; the structure is shown in Figure 9. If the di_format field of the inode core is 2, then the inode data fork consists of a structure of data extents.

| | 0 | 1 | 2 | 3 | 4 | 5 | 6 | 7 | 8 | 9 | A | B | C | D | E | F |
|---|---|---|---|---|---|---|---|---|---|---|---|---|---|---|---|---|
| 0x00 | Magic Number (0x58465342) | | | | Block_Size | | | | Block Count | | | | | | | |
| 0x10 | Blocks(used Real-time device) | | | | | | | | Extents(used Real-time device) | | | | | | | |
| 0x20 | UUID | | | | | | | | | | | | | | | |
| 0x30 | Journaling Log start block(offset) | | | | | | | | Root Inode # | | | | | | | |
| 0x40 | Bitmap Inode | | | | | | | | Real-time Bitmap Inode | | | | | | | |
| 0x50 | Extent size(in Real-time device) | | | | AG Block Size | | | | AG Count | | | | Bitmap Block cnt(in real-time device) | | | |
| 0x60 | Log Block cnt | | | | Ver. # | | Sector Size | | Inode Size | | Inode Per Block | | File System Name[12] | | | |
| 0x70 | File System Name[12] | | | | | | | | Block Log | Sector Log | Inode Size Log | Inopblk Log | AGblk Log | Rextents Log | In Progress | Imaxpct |
| 0x80 | Inode Count | | | | | | | | Free Inode Count | | | | | | | |
| 0x90 | Free Block Count | | | | | | | | Free Extent Count | | | | | | | |
| 0xA0 | uquotino | | | | gquotino | | | | qflags | | flag | share vn | inoalignment | | | |
| 0xB0 | unit | | | | width | | | | Dirblk log | Logsect log | Logsec size | | Log sunit | | | |
| 0xC0 | features2 | | | | bad features2 | | | | features compat | | | | features ro compat | | | |
| 0xD0 | features log incompat | | | | crc | | | | spino align | | | | pquotino | | | |
| 0xE0 | pquotino | | | | lsn | | | | meta uuid | | | | | | | |
| 0xF0 | meta uuid | | | | | | | | rrmapino | | | | | | | |

**Figure 5.** Superblock structure of XFS file system.

| AG Number | # bits = agblklog | # bits = inopblklog |
|---|---|---|

MSB                                  LSB

**Figure 6.** Inode number structure of XFS file system.

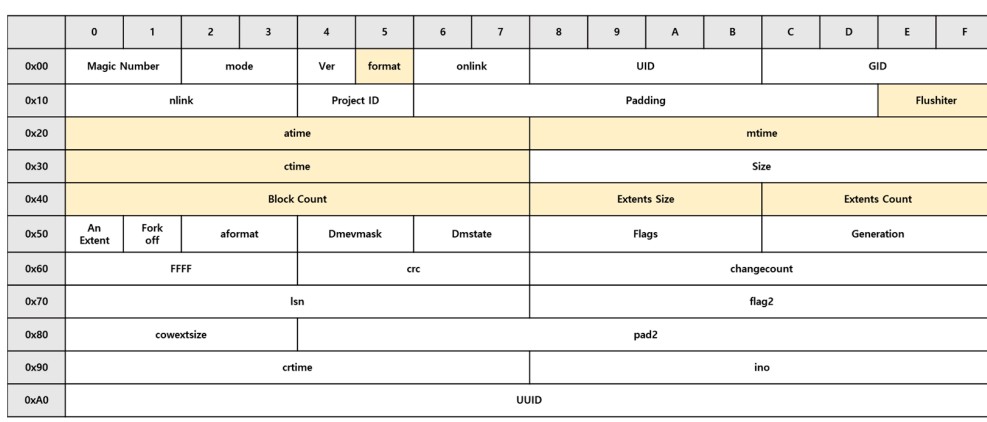

**Figure 7.** Inode core structure of XFS file system.

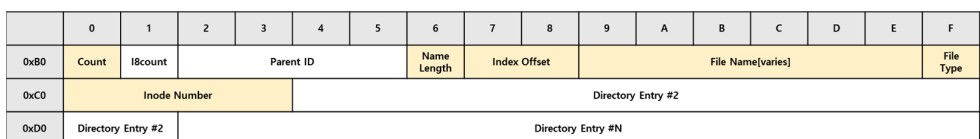

**Figure 8.** Directory entries structure of XFS file system.

| Flag (1) | Allocation Group Number (54) | Allocation Block Number agblklog (52) | Block Count (21) |
|---|---|---|---|

**Figure 9.** Structure of data extents of XFS file system.

### 3.2.2. XFS v5-Based File Extraction Method

The file extraction framework using the metadata of XFS v5 proposed in this paper is shown in the Figure 10. The proposed framework searches for the general information of the file system in the superblock: root inode number, sector size, block size, inode size, InopblkLog, and agblkLog. The root inode number is used to find the address of the first

inode, and InoblkLog and agblkLog are utilized for calculating the allocation group, block, and sector to locate the address of the inode to which the actual file is allocated.

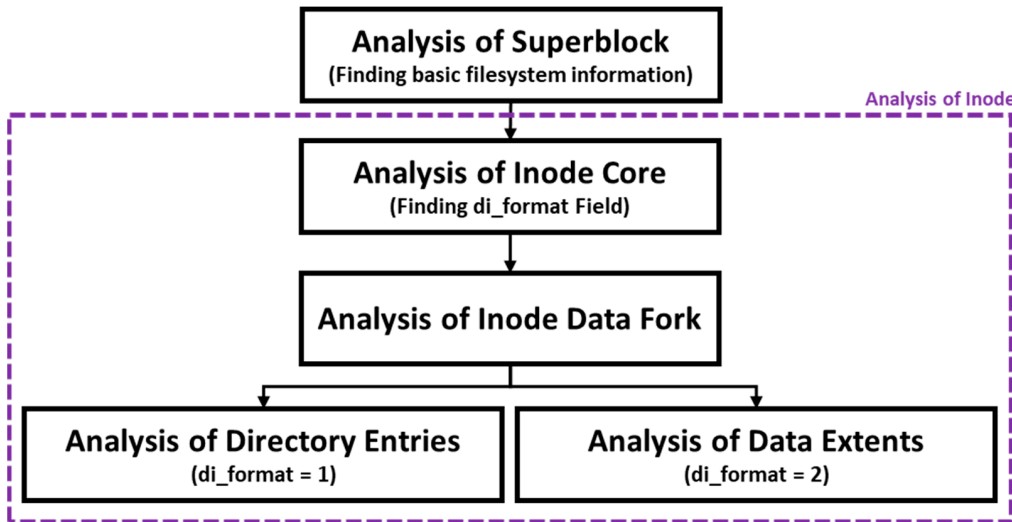

**Figure 10.** XFS v5-based file extraction module using TSK.

Thereafter, it finds the address of the inode and that of the actual file corresponding to the inode. The inode address can be located by calculating the allocation group, block, and sector values using the inode number, agblklog, and inopblklog, as shown in Figure 3. The file attribute value can be determined by the inode core, and the address of the actual file can be found by analyzing the inode data fork structure to match the value of the di_format field in the inode core. If the di_format is 1, then the file name and inode offset are identified through directory entry structure analysis. If the di_format is 2, then the address where data are stored can be obtained through the bitwise operation of the structure of data extents, and the allocated file can be extracted.

*3.3. Comparison of Previous Research and Proposed Framework*

The comparison of the TSK-based file extraction and recovery framework proposed in this paper with the existing studies based on five criteria is shown Table 1. Verification refers to hash comparison to ascertain the integrity of the data corresponding to the file. Fairbanks et al. [5] and Lee et al. [6] proposed a file recovery method considering that the deleted inode is stored in the journal area. Andreas Dewald et al. [13] proposed a file recovery method for the Ext4 file system through a pattern-based carving in which actual files are assigned to the inode. However, the metadata analysis for file recovery is insufficient, and determining whether it has been normally extracted and recovered is difficult because hash value comparisons between the original and recovered files are not performed. Park et al. [7] and Ahn et al. [8] also proposed a file recovery method considering that the deleted inode is stored in the journal area, and the extent indicating the location of the actual data in the file is not deleted. However, the metadata analysis used for file recovery is insufficient. The method encounters a problem in determining whether the file has been extracted and recovered normally because the hash value comparisons between the original and recovered files are not performed. Most existing studies insufficiently describe the metadata used by the actual recovery method. Moreover, they do not compare the hash values between the original and recovered files; the file system and environment are not specified. Accordingly, this paper elaborates on the metadata of the file system related to the extraction and recovery of files, supplementing the insufficient parts in previous studies. In addition, the applicability of the proposed file extraction and recovery method as well as the file system version are clearly explained.

**Table 1.** Comparison of existing file extraction and recovery research with the proposed framework.

|  | Metadata Analysis | Recovery Method | Verification | Tool | Environment |
|---|---|---|---|---|---|
| Fairbanks et al. [5] (2012, Ext4) | - Inode<br>- Extent<br>- Journal Data block | Journal Area | Not Supported | Not Supported | Ubuntu 10.04 (2.6.32-38 kernel) |
| Lee et al. [6] (2019, Ext4) | - Inode Bitmap<br>- Inode<br>- Journal Data block | Journal Area | Not Supported | Not Supported | Ubuntu 14.04 |
| Andreas Dewald et al. [14] (2017, Ext4) | - Inode<br>- Extent | Carving | Not Supported | Not Supported | Not specified |
| Ahn et al. [7] (2014, XFS) | - Superblock<br>- Inode Core<br>- Inode Data Fork (formats 1–3) | Journal Area | Not Supported | Supported | Slackware (3.1.0 Kernel) |
| Park et al. [8] (2016, XFS) | - Superblock<br>- Inode Core<br>- Inode Data Fork (formats 1–2) | Journal Area | Not Supported | Supported | Not specified |
| Zheng Wang et al. [15] (2016, XFS) | - Journal Area<br>- Inode Core<br>- Inode Data Fork (format 2) | Not Supported | Not Supported | Not Supported | Not specified |
| Proposed Framework (Ext4) | - Journal Superblock<br>- Journal Descriptor Block<br>- Journal Data Block | Journal Area | Supported | Supported | Ubuntu 20.04 (5.4.0-47-generic) |
| Proposed Framework (XFS) | - Superblock<br>- Inode Core<br>- Inode Data Fork (formats 1–2) | Inode | Supported | Supported | Ubuntu 20.04 (5.4.0-47-generic) |

## 4. Proposed Framework Validation Experiment and Results

### 4.1. Validation Experiment Environment and Dataset

To verify the TSK-based Ext4 file system data recovery and XFS file system file extraction framework proposed in this paper, 1 and 100 GB partition datasets were constructed for each file system. In addition, to verify the extraction and recovery feature of the file system, the experiment was conducted by dividing the images into images before and after data deletion. The verification environment is listed in Table 2. In the XFS file system dataset, version 5.3.0 of xfsprogs was used to create XFS v5. In the Ext4 file system dataset, version 1.45.5 of e2fsprogs built in Ubuntu 20.04 was employed [16].

**Table 2.** Experimental environment.

| Environment | Version |
|---|---|
| OS | Ubuntu 20.04 (Bionic Beaver) |
| TSK | 4.6.4 |
| xfsprogs | 5.3.0 |
| e2fsprogs | 1.45.5 |

The constructed dataset summarized in Table 3 includes the jpg, pptx, docx, and pdf file formats; each format has 25 files. The 1G_after and 100G_after datasets are the deletion of 12 files by the file type from 1G_before and 100G_before. In the case of file deletion, 12 files, which represent half of the total number of files, have been deleted in order to easily compare the recovery rate and accuracy used in the extraction and recovery performance evaluation.

**Table 3.** Number of files/total number of files by file type that exist within the dataset used in the experiment.

| Dataset | PDF | JPG | DOCX | PPTX |
|---|---|---|---|---|
| 1 GB_before | 25/100 | 25/100 | 25/100 | 25/100 |
| 1 GB_after | 13/52 | 13/52 | 13/52 | 13/52 |
| 100 GB_before | 25/100 | 25/100 | 25/100 | 25/100 |
| 100 GB_after | 13/52 | 13/52 | 13/52 | 13/52 |

*4.2. Validation Result*

Tables 4 and 5 list the experimental results for the Ext4 and XFS datasets based on the recovery rate and accuracy as recovery and extraction success indicators. The recovery rate refers to the percentage of the total number of existing files in a dataset and the number of deleted files that have been recovered. Accuracy refers to the percentage of the exact number of existing files and the number of deleted files. Specifically, in this study, we determined the exact recovery criteria for a file by verifying that the hash values of the files completely matched.

**Table 4.** Experimental result based on recovery rate and accuracy for Ext4.

| Dataset | TSK | | Proposed Framework | | UFS Explorer | | R-Studio | |
|---|---|---|---|---|---|---|---|---|
| | REC | ACY | REC | ACY | REC | ACY | REC | RCY |
| 1GB_before | 100 | 100 | 100 | 100 | 100 | 100 | 100 | 100 |
| 1GB_after | 100 | 100 | 85 | 77 | 56 | 56 | 100 | 52 |
| 100GB_before | 79 | 79 | 100 | 100 | 100 | 100 | 100 | 100 |
| 100GB_after | 53 | 50 | 100 | 79 | 88 | 74 | 99 | 50 |

**Table 5.** Experimental result based on recovery rate and accuracy for XFS-v5.

| Dataset | TSK | | Proposed Framework | | UFS Explorer | | R-Studio | |
|---|---|---|---|---|---|---|---|---|
| | REC | ACY | REC | ACY | REC | ACY | REC | RCY |
| 1 GB_before | - | - | 100 | 100 | 100 | 0 * | 94 | 0 ** |
| 1 GB_after | - | - | 52 | 52 | 52 | 0 * | 94 | 0 ** |
| 100 GB_before | - | - | 100 | 100 | 100 | 0 * | 97 | 0 ** |
| 100 GB_after | - | - | 52 | 52 | 52 | 0 * | 91 | 0 ** |

\* hash value is not matched. \*\* hash value and file name are not matched.

In the Ext4 file system, there are cases where metadata information is not backed up in the journal area; consequently, the recovery rates are 85% and 100% for the 1 GB and 100 GB partitions, respectively. However, accuracy tends to decrease because there are also cases in which the area where the actual data are stored is overwritten or the backed-up inode is not the most recent file. In the case of the XFS file system, the recovery rate of the 1 GB and 100 GB partitions was 52%. When a file is deleted from the XFS file system, all metadata related to the file are initialized to 0, making recovery impossible.

**5. Discussion**

In this study, we analyzed the metadata changes in Ext4 journal checksum v3 and XFS v5. Based on the analysis, we found that the structure of the descriptor block in the Ext4 file system has been changed according to the value of the s_feature_incompat field of the journal superblock in the journal area. We further discovered that XFS v5 added new fields to guarantee integrity (e.g., uuid and crc) in metadata structures, such as the inode and superblock. Using this, a file extraction and recovery framework based on TSK is proposed. The dataset was used to compare the recovery rate and accuracy of the proposed framework with existing commercial tools. The dataset was constructed by

storing 25 files in each directory corresponding to the file extension and deleting 12 files each by file extension. The experimental results indicated that the recovery function using the journal area of the Ext4 file system was not implemented in the original TSK, and the recovery rate and accuracy could not be measured because the XFS file system was not a supported file system. In UFS Explorer, in the case of Ext4 Journal Descriptor v3, the file stored in the directory could not be restored. In the case of XFS v5, the actual file could not be restored because it was restored to a file with a data size of 0. In R-Studio, in the case of Ext4 Journal Descriptor v3, most of the files were recovered, but the accuracy was low due to different hash values. In the case of XFS v5, most of the files were recovered, but the hash value and file name did not match. As for the proposed framework, file recovery was performed based on the metadata of the file system, and its recovery rate and accuracy were relatively high compared with those of commercial tools. However, in the case of the XFS file system, the proposed technique did not consider the Btree inode and journal area. Thus, it was impossible to recover files with a file size of 100 GB. We intend to rectify this and consider implementing the recovery function using the journal area of the Ext4 file system as well as applying the file extraction and recovery function of the XFS file system to the open source of TSK.

## 6. Conclusions

File systems are constantly updated, and major metadata changes that are implemented can affect the success or failure of digital forensic investigations. Accordingly, a technique for analyzing the metadata change in the Ext4 and XFS file systems according to this necessity was proposed, and a TSK-based framework was developed. In addition, we compared its recovery rate and accuracy with those of existing commercial tools and verified that it was more effective. In particular, by applying the file extraction method of the XFS file system to the proposed framework, it was proven that unsupported file systems can be extended.

As such, it was confirmed that it can be helpful in file system forensic investigations by presenting a file extraction and recovery methodology through metadata analysis of the file system before and after data deletion. Therefore, in future research, forensic research targeting the file system environment applied to actual CCTV and servers, and metadata analysis and recovery methodologies for Btrfs and OpenZFS, should be performed. In addition, research on a TSK-based integrated file system forensic framework that can be used by real investigators should be considered.

**Author Contributions:** Conceptualization, H.K., S.K., Y.S. and T.S.; methodology, H.K., S.K., Y.S., W.J. and S.L.; validation, H.K., S.K. and Y.S.; formal analysis, H.K., S.K., Y.S., W.J. and S.L.; investigation, H.K.; software, H.K., Y.S. and W.J.; writing—original draft preparation, H.K.; writing—review and editing, H.K., S.K., Y.S., W.J., S.L. and T.S. All authors have read and agreed to the published version of the manuscript.

**Funding:** This research received no external funding.

**Acknowledgments:** This research was supported by the Basic Science Research Program through the National Research Foundation of Korea (NRF) funded by the Ministry of Science, ICT & Future Planning (NRF-2018R1D1A1B07043349) and the Energy Cloud R&D Program through the National Research Foundation of Korea (NRF) grant funded by the Ministry of Science, ICT (2019M3F2A1073385).

**Conflicts of Interest:** The authors declare no conflict of interest.

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
