# Peer review of "Ext4 and XFS File System Forensic Framework Based on TSK"

_electronics, doi:10.3390/electronics10182310_

Round 1
Reviewer 1 Report
In this manuscript, a framework for extracting and recovering files based on The Sleuth Kit (TSK) is proposed by deriving the metadata changed in Ext4 file system journal checksum v3 and XFS file system v5. Moreover, by comparing the accuracy and recovery rate of the proposed framework with existing commercial tools using the experimental dataset, the authors found that sustained research on file systems should be conducted from the perspective of forensics. To the best of my knowledge, both the proposed file system forensic framework and the experiment result are correct. Indeed, I found the idea is interesting. Thus, I can recommend this manuscript for publication in Electronics.
Reviewer 2 Report
Formatting:
Title Case should be carefully applied to subtitles as well.
Text organization:
Table 10. with comparison of related work to the proposed approach should be provided in section 3 with the proposed file system framework, not in section 4 (experiment and results)
The Manuscript Content:
Title of the paper announces forensic framework, but the proposed method and results are related to extraction and recovery of deleted files. This, in fact, could be used in forensics, but it should not be emphasized in title, since the forensics case study has not been presented in results. Therefore, it is necessary to adjust title and abstract with the results. It is also important to have the complete manuscript, especially section 3 (the proposed method) adjusted with the results. If authors would like to keep the title (having forensics emphasized), then forensics methods should be mentioned in theoretical background, related work and results should be provided for some forensics case.
Results section:
Table 7. presents "current files" vs total number of files in the experiment. The term "current" is not clear.
Conclusion section:
Too short. Maybe to elaborate more the results and possible research directions.
Reviewer 3 Report
The reason why I am rejecting this paper is because I do not see how it fits to the scope of the journal (electronics) or the special issue on artificial intelligence applications.
I suggest re-submission to an appropriate conference or journal.
The second problem is that the paper reads more as a technical report on a software solution rather than a research paper.
It is also not very clear from the paper how the analysis is done, what if data is partially corrupted, which is often the case? Is partial recovery even possible? In my opinion the paper should focus on the analysis and its steps rather than details of the structure of the file systems. If the analysis is a simple match between the expected structure and what is found on the device it might not be sufficient.
I have also added a few comments on the paper organization to help with any future submission:
Although recovery of delete files is important a scientific paper should abstain in making empty claims such as: "In an actual forensic investigation, a deleted file that cannot be recovered can lead to profound consequences".
A subtitle should not be placed immediately after the title, some introductory text should be added (this is the case for section 2 and subsection 2.1 at least).
The related work is written as a historical paper, focusing on the years when the papers were published instead of focusing on the work and how it relates to what is presented in this paper.
Table 1 is not really a table it is a code outline. It should be marked as such. This should also be done for others. The bigger problem is that the code outline is a particularity of the file system and not invented or developed by the authors, as such it should not be added at all but instead the proper references should be made.
The font size of figure 2 is extremely large compared to everything else.
Table 10 is too large and extends beyond the borders.
Refrences are formatted differently from the rest of the text.
Reviewer 4 Report
Dear Authors,
I give you my extended remarks and suggestions regarding the article "Ext4 and XFS File System Forensic Framework based on TSK".
A few remarks and suggestions:
- The article is solid structured from summary to conclusion and the whole content makes one meaningful whole without deviating from the main topic,
- The research covered in the article bring contributions as follows:
-
- The characteristics of the journal area used for the recovery of deleted data in the fourth extended file system and those of the metadata used for data extraction in the XFS file system are derived,
-
- A file system forensic tool capable of recovering deleted data from the fourth extended file system and extracting XFS file system data based on The Sleuth Kit (TSK) using the characteristics of each derived file system is developed,
-
- The developed TSK-based file system forensic tool can be applied to large-capacity environments exceeding 100 GB,
- The introduction gives a concise overview of processed scientific field,
- Also, introduction indicates the area that the researchers detected, and in which they have contributed concrete scientific contributions,
- Notes:
-
- I suggest another review of the English translation,
-
- There are a lot of abbreviations of various terms throughout the article, which is quite normal for the topic, but I ask that abbreviations for ALL terms be defined when they first appear in the text (rules for writing articles and rules for Journal Electronics),
-
- In the selected methodology, The Sleuth Kit was chosen, I suggest that the authors explain why this tool, which has its limitations (which the authors shyly mention) and not some other (perhaps through a comparative analysis of the tools, can also be presented in tabular form),
-
- I suggest that the article be further strengthened with relevant and newer sources of literature (not excluding and not devaluing the current literature),
-
- The types of simulated files mentioned in the experiment are mentioned but apart from the summarized results no results are mentioned by file type, consider adding a tabular view of these results,
-
- It is mentioned in the introduction of the article the types of devices that use the analyzed file systems, whether this experiment is designed to perform forensic analysis on some "real" existing device and if so which (perhaps add to the article),
- The article gives interesting results and a good introduction to future research with real data on real devices, and implementation of results in real scenarios,
- The article contributes with concrete scientific contribution,
- The article contributes to the Scientific Journal Electronics,
- My assessment is that the article be accept after minor revision for publication in a journal.
My concluding opinion is that the article is accepted after minor revision for publication!
Sincerely
Round 2
Reviewer 2 Report
Authors made corrections according to the review comments. The paper is significantly improved - it is more clear and precise.
The paper is now appropriate for publication.
NOTE: Bottom of page 9 contains only table caption, while the table content is put at the following page 10. It is necessary to finalize paper to have the whole table at the same page, e.g. page 10.
